# Exosome-Derived microRNAs from Mouthrinse Have the Potential to Be Novel Biomarkers for Sjögren Syndrome

**DOI:** 10.3390/jpm12091483

**Published:** 2022-09-10

**Authors:** Kouta Yamashiro, Tomofumi Hamada, Kazuki Mori, Keitaro Nishi, Maya Nakamura, Mahiro Beppu, Akihiko Tanaka, Hiroshi Hijioka, Yoshiaki Kamikawa, Tsuyoshi Sugiura

**Affiliations:** 1Department of Maxillofacial Diagnostic and Surgical Science, Field of Oral and Maxillofacial Rehabilitation, Graduate School of Medical and Dental Science, Kagoshima University, Kagoshima 890-8580, Japan; 2Department of Oral & Maxillofacial Surgery, Hakuaikai Medical Cooperation, Sagara Hospital, Kagoshima 892-0833, Japan

**Keywords:** Sjögren syndrome, mouthrinse, microRNA, exosome, liquid biopsy, diagnosis, screening test, oral science

## Abstract

Sjögren syndrome (SS) is diagnosed based on invasive tissue biopsies and blood sampling. Therefore, a novel non-invasive and simple inspection diagnostic marker of SS is required. Here, we identified exosome-derived microRNAs (miRNAs) as biomarkers for SS using non-invasive mouthrinse samples collected from patients with SS and healthy volunteers. We compared miRNAs derived from exosomes in mouthrinse samples from the two groups using microarrays and real-time polymerase chain reaction (PCR) and identified 12 miRNAs as biomarker candidates. The expression ratios of four miRNAs were significantly increased in the SS group compared to the control group. Logistic regression analysis revealed a more significant influence of miR-1290 and let-7b-5p in the SS group than that in the control group. We combined these miRNAs to create a diagnostic prediction formula using logistic regression analysis. The combination of miR-1290 and let-7b-5p distinguished SS from the control samples with an AUC, sensitivity, specificity, positive predictive value, and negative predictive value of 0.856, 91.7%, 83.3%, 84.6%, and 90.9%, respectively. These results indicated that an increased ratio of these miRNAs could serve as a novel and non-invasive diagnostic marker for SS. This is the first report of diagnosis and screening of SS by adopting a non-invasive method using mouthrinse.

## 1. Introduction

Sjögren syndrome (SS) is an autoimmune disease characterized by a dry mouth and eyes [1,2]. It comprises primary SS, which is not associated with other autoimmune diseases, and secondary SS, which is associated with other autoimmune states such as rheumatoid arthritis and systemic lupus erythematosus [3]. Presently, SS is diagnosed in Japan using the 1999 revised Japanese Ministry of Health and Welfare criteria. The 2002 American–European Consensus Group (AECG) Classification Criteria for SS [4], the 2012 American College of Rheumatology (ACR) Classification Criteria for SS [5], and the 2016 American College of Rheumatology/European League Against Rheumatology (ACR/EULAR) Classification Criteria for primary SS [6] are applied internationally. However, these diagnostic strategies involve the invasive and complicated collection of tissue biopsy and blood samples and imaging procedures using radiopharmaceuticals and contrast agents, which are burdensome for patients [7]. Although anti-SS-A and anti-SS-B antibodies obtained by invasive blood sampling are included in these criteria [8], the detection rates of primary and secondary SS are only 33–74% and 23–52%, respectively [9]. Therefore, simple, non-invasive examinations and more precise identification of biomarkers are required.

MicroRNAs (miRNAs) are single-stranded RNA molecules of 18–25 bases that do not encode proteins but regulate gene expression [10,11,12]. They are associated with important functions in various biological activities, such as cell proliferation, differentiation, apoptosis, ontogeny, immunoregulation, aging, and carcinogenesis [11,13]. MicroRNAs are found in the saliva and urine, as well as the blood, and might serve as new biomarkers for malignant tumors and autoimmune diseases [14,15]. These miRNAs are secreted from cells via exosomes, which are 40–150 nm vesicles [16,17] that, during secretion, protect the miRNAs from being digested by enzymes in the body [18]. Accordingly, microRNAs are exchanged between cells, and they might regulate gene expression [15,19]. Since miRNAs secreted from intracellular vesicles can be absorbed by cells of body fluids, those derived from exosomes in body fluids could serve as biomarkers of diseases [16,17,18,19].

Exosome-derived miRNAs and proteins in salivary gland tissue, blood, and saliva might be useful as diagnostic biomarkers for SS [10,13,20,21]. However, tissue biopsy and blood collection from patients are invasive. Moreover, patients with SS often produce little saliva, making it difficult to collect amounts sufficient for analysis. Therefore, we non-invasively rinsed the oral cavity with a specific volume of fluid to collect sufficient amounts of secreted saliva and exfoliated oral cells for analysis. The oral florae in saliva and mouthrinse are similar and closely correlated [22]. If mouthrinse can be used in place of saliva, samples can be easily collected, even from patients with SS who produce low saliva volumes and have xerostomia. However, the relationship between exosome-derived miRNAs in mouthrinse and SS remains unknown. Therefore, in this study, we identify exosome-derived miRNAs in mouthrinse and substantiate their value for the non-invasive diagnosis of SS. 

## 2. Materials and Methods

### 2.1. Study Design and Participants

This study included 24 female patients with SS who were treated at the Department of Oral Surgery, Kagoshima University Hospital Oral Maxillofacial Center (Kagoshima, Japan) between 2019 and 2020 and 24 healthy volunteers (controls). The patients were diagnosed with primary SS based on meeting two or more of the four criteria in the 1999 revised Japanese Ministry of Health and Welfare criteria for the diagnosis of SS, the 2002 AECG Classification Criteria for SS, the ACR Classification Criteria for SS, and the 2016 ACR/EULAR Classification Criteria for primary SS. Patients with a dry mouth, malignant, inflammatory, or autoimmune diseases were excluded.

### 2.2. Sample Collection

Mouthrinse samples were collected in the morning between 9 am and noon. The patients did not have symptoms of fever or colds and were generally well on collection days. The participants refrained from smoking, eating, drinking, chewing gum, and brushing their teeth for 30 min before sampling. The patients sat upright in a chair. They sipped 10 mL of sterile purified water and rinsed this evenly in their mouths for 1 min. The patients were then instructed to drain the entire mouthrinse into a sterile container. Collected mouthrinse samples were stored at −80 °C until analysis.

### 2.3. Exosome Isolation

Exosome pellets (150 μL) were prepared at 4 °C from mouthrinse samples (1.5 mL) using an ultracentrifugation protocol [23] with some modifications. Briefly, mouthrinse samples (1.5 mL) were centrifuged at 300× *g* for 10 min and then the first supernatant was centrifuged at 2000× *g* for 10 min. The second supernatant was centrifuged at 10,000× *g* for 30 min and then the third supernatant (3) was balanced in 2.2-mL ultracentrifugation tubes (Beckman Coulter Inc., Brea, CA, USA) using phosphate-buffered saline (PBS) and centrifuged again at 100,000× *g* for 70 min in a TLS-55 Swinging-Bucket rotor (346134; Beckman Coulter, Inc., Brea, CA, USA). The final supernatant was removed, and the exosomes were precipitated, washed with PBS, and centrifuged again at 100,000× *g* for 70 min. The supernatant was discarded, and exosomes were resuspended in a final volume of 150 μL of PBS. Isolated exosome particles were confirmed by nanoparticle tracking analysis (NTA) using NanoSight LM10 (Malvern Panalytical Ltd., Malvern, Worcestershire, UK) [24,25] and then reconfirmed by Western blotting [25] as described previously [26]. Specifically, samples were resolved via sodium dodecyl sulfate polyacrylamide gel electrophoresis at 100 V for 60 min using 4–15% mini-PROTEAN^®^ TGX^TM^ gels (Bio-Rad Laboratories Inc. Berkeley, CA, USA) and transferred to Trans-Blot membranes (Bio-Rad). Nonspecific protein binding was blocked using Blocking One solution (NacalaiTesque, Kyoto, Japan) and then the membranes were incubated with primary anti-human CD9 (diluted 1:2000) and anti-human CD63 (diluted 1:1000) antibodies for 1 h, washed, and incubated with secondary goat anti-Mouse IgG antibody (diluted 1:5000). Thereafter, the chemiluminescence was detected using Immunostar LD (Fuji Film Wako, Osaka, Japan).

### 2.4. Isolation of RNA 

Small RNAs isolated from exosome pellets (150 μL) using miRNeasy Serum/Plasma Mini Kits (Qiagen, GmbH, Hilden, Germany) as described by the manufacturer were directly applied to miRNA microarray and real-time polymerase chain reaction (real-time PCR) analyses.

### 2.5. MicroRNA Microarray

We analyzed differentially expressed miRNAs in pooled mouthrinse samples from ten randomly selected patients with SS and ten controls. Pooled mouthrinse RNA was labeled using 3D-Gene-miRNA labeling kits (Toray Industries Inc., Kamakura, Japan) as described by the manufacturer and then hybridized to 3D-Gene miRNA array platform ver.21 comprising 2633 miRNAs (Toray Industries Inc., Kamakura, Japan). After stringent washing, fluorescence signals were visualized using a 3D-Gene Scanner 3000 (Toray Industries Inc, Kamakura, Japan) and analyzed using 3D-Gene Extraction software (Toray Industries Inc, Kamakura, Japan).

Raw data for each spot were normalized by replacing the average strength of the background signal, representing the signal intensities of all blank spots, with 95% confidence intervals (CI). Spots with signal intensities >2 standard deviations (SD) above the background signal intensity were considered valid. The relative expression of miRNAs was calculated by comparing the signal intensity of valid spots throughout the microarray. The data were globally normalized on a per-array basis such that the median signal intensity was adjusted to 25. Relative hybridization intensities and background hybridization values were calculated. Significant changes were defined as >2.0- or <0.5-fold by comparing the results of the SS and control groups.

### 2.6. Real-Time PCR

The expression of exosome-derived miRNAs in mouthrinse samples was compared using quantitative real-time PCR, TaqMan^®^ MicroRNA Assays (Applied Biosystems, Foster City, CA, USA), and TaqMan^®^ Fast Advanced Master Mix (Applied Biosystems, Foster City, CA, USA), with specific primers for 15 miRNAs, as described by the manufacturer. The reverse transcription conditions comprised incubating samples at 16 °C for 30 min, 42 °C for 30 min, 85 °C for 5 min, then 4 °C. The products were stored at −20 °C. The PCR cycling conditions were as follows: 10 min at 95 °C, followed by 45 cycles of 15 s at 95 °C, and 60 s at 60 °C. The cycle threshold (Ct) value was determined using a background-based threshold calculated by Step One Software ver.2.3 (Applied Biosystems, Foster City, CA, USA). Due to the lack of an internal control for miRNAs [27], we selected miR-24 and miR-155-5p as endogenous internal controls. This decision was made since others have applied miR-24 as an endogenous control [27], and the expression of miR-155-5p did not differ between the SS and control samples in the miRNA microarray. The expression of candidate control miRNAs was analyzed using real-time-PCR (data not shown), and the mean values for stable miR-24 and miR-155-5p expression were selected as the endogenous control. The relative expression of target miRNA to the control in each sample was calculated using the 2-ddCt method as:ddCt patient = Ct patient − (mean Ct of miR-24 and miR-155-5p) − dCt healthy control
dCt healthy control = mean Ct of healthy control (Ct mean value of miR-24 and miR-155-5p)

### 2.7. Statistical Analysis

Differences in clinical characteristics between the SS and control groups were evaluated using Fisher’s exact, Student’s t, and Mann–Whitney U tests. The relative expression of miRNAs in the SS and control samples was compared using Student’s t and the Mann–Whitney U tests. The diagnostic performance of candidate miRNAs was determined using the area under the receiver operating characteristics (ROC) curve (AUC). Associations between miRNAs and SS were evaluated using cut-off values determined from the AUCs and Fisher’s exact tests. A logistic regression prediction model was applied to candidate miRNAs to calculate prediction scores for individual samples. Whether the data of any two groups were normally distributed was confirmed using Shapiro–Wilk tests. The homoscedasticity of normally distributed data was confirmed using Levene’s tests, and then the appropriate test was applied. Statistical analyses were performed using SPSS software ver.26 (IBM Co., Armonk, NY, USA). Values with *p* < 0.05 were deemed statistically significant.

## 3. Results

### 3.1. Clinical Characteristics of Study Participants

Table 1 shows the clinical pathological characteristics of the 24 female patients with SS (median age 63.8 (35–86) years) and 24 healthy female controls (median age 63.5 (41–84) years). The age and proportions of smokers did not significantly differ between the two groups. The average duration of SS was 88 (13–204) months and the positive rates of anti-SS-A and SS-B antibodies were 75% and 42%, respectively. The average amount of saliva obtained from patients with SS in the Saxon test was 1.96 (0.2–6.1) g. Muscarinic acetylcholine receptor M3 (M3R) inhibitors and corticosteroids were, respectively, found in 15 (62.5%) and 3 (12.5%) patients with SS.

### 3.2. Preparation of Exosome Pellets

No study previously described the isolation of exosomes from mouthrinse using ultracentrifugation. The maximum and average sizes of the fine particles of pellets determined by NTA were 106 and 145 nm, respectively (Figure 1a). Microvesicles, that is, presumed exosomes, comprised most of the particles. Furthermore, Western blotting of pellet extracts from mouthrinse samples revealed the exosome markers CD9 and CD63 (Figure 1b).

### 3.3. Comprehensive Analysis of Exosome-Derived miRNAs 

We analyzed candidate miRNAs for SS diagnosis using miRNA microarrays (Figure 2a). Mouthrinse exosomes from 10 randomly selected patients with SS were pooled and the RNA profiles were compared with those of pooled control samples. Among the 2633 miRNAs, 241 were upregulated >2-fold and 161 were downregulated <0.5-fold in the SS sample. From a total of 402 miRNAs, 12 were selected (let-7b-5p, miR-1290, miR-29b-2-5p, miR-3124-5p, miR-3200-5p, miR-34a-5p, miR-3648, miR-371b-5p, miR-4787-5p, miR-5100, miR-512-3p, and miR-640; Figure 2b), which showed a high rate of change in expression levels, or for which there were literature reports associated with SS [12,20].

### 3.4. Verification of 12 Candidate miRNAs Using Real-Time PCR

A total of 48 samples (24 SS patients and 24 healthy controls) were verified through real-time PCR of the 12 candidate miRNAs. The results showed significantly increased amounts of let-7b-5p, miR-1290, miR-34a-5p, and miR-3648 in exosomes derived from mouthrinse of patients with SS than that from controls (*t*-test, *p* < 0.05; Figure 3a and Appendix A). These miRNAs were also upregulated in the SS group according to the microarray analysis. The miRNAs miR-3200-5p, miR-3124-5p, miR-4787-5p, miR-5100, and miR-512-3p did not significantly differ between the SS and control groups (Appendix A). We excluded miR-640, miR-29b-2-5p, and miR-371b-5p from further investigation due to their low average levels (Ct > 40) and unstable expression in >50% of the participants.

### 3.5. Ability of One miRNA to Diagnose SS

We determined signal intensity cut-offs for the nine remaining miRNAs using the AUC (Figure 3b and Appendix A). Table 2 shows the AUC, sensitivity, specificity, positive (PPV) and negative (NPV) predictive values, and Fisher’s exact test values for SS detection. Based on these cut-off values, among the nine miRNAs, let-7b-5p, miR-1290, and miR-34a-5p significantly differed between the SS and control group (*p* < 0.05, Fisher’s exact tests; Table 2). The AUC of these three miRNAs was >0.7, suggesting that all of them could detect and diagnose SS (Figure 3b).

### 3.6. Combining Two miRNAs Accurately Diagnosed SS 

Figure 4a shows the effects of combining miRNAs on the accuracy of SS detection, determined using logistic regression analysis. The diagnostic miRNA index of the miR-1290 and let-7b-5p combination, calculated using the following regression equation,
SS diagnostic miRNA index = 1 − 1/ [1 + exp (−1 × score)]Score = 86.981 + (1.883 × let-7b-5p) + (0.916 × miR-1290),
accurately diagnosed SS (AUC: 0.856, *p* < 0.001; Figure 4b,c). The sensitivity, specificity, PPV, and NPV were 91.7%, 83.3%, 84.6%, and 90.9%, respectively, when the cut-off obtained from ROC curves was the diagnostic index (Figure 4d), also indicating high accuracy. This miRNA combination accurately distinguished the SS group from the control group and could predict SS with statistical significance. The combination had higher diagnostic power than either miRNA alone.

## 4. Discussion

This study examined biomarkers with which to diagnose SS. The expression of several miRNAs useful for diagnosing SS differed between SS and control groups [12,20,28,29]. Sampling minor salivary glands, salivary gland epithelial cells, and peripheral blood mononuclear cells is diagnostically reliable but stressful for patients. Therefore, we developed a simple, accurate, and stress-free method of sampling using mouthrinse.

Many patients with SS have extremely dry mouths due to decreased salivation. Our preliminary study showed that collecting sufficient volumes of samples for exosome extraction was onerous. In addition, the purity and quantity of exosomes isolated from such saliva samples were unstable. To overcome those limitations, we rinsed the oral cavity with water and collected residual saliva and secretions. Prolonged, repeated rounds of high-speed ultracentrifugation removed any remaining mucosal cells and bacteria and allowed for the collection of saliva- and cell-derived exosomes. This safe and straightforward approach to sample collection can easily be applied to screen diseases [22]. Furthermore, repeated sample collection should facilitate disease progress and severity monitoring. To the best of our knowledge, using mouthrinse for SS diagnosis has not been previously reported, and hence, this approach has not been applied in clinical practice. We believe that using mouthrinse to collect SS biomarker samples holds promise as a non-invasive method for the diagnosis of SS.

Exosomes contain not only miRNAs but also various types of biological information such as information on genes and proteins. Our goal was to identify SS-specific miRNAs from exosomes contained in discharged mouthrinse. Therefore, screening of miRNAs from mouthrinse-derived exosomes was first considered. Screening was performed using pooled mouthrinse samples from 10 SS patients and 10 healthy controls using a miRNA microarray platform according to a previously described method [30]. We did not use the procedure to verify all of the 402 miRNAs because of the cost and time involved. Instead, 12 miRNAs were selected based on the increase/decrease in the expression ratio in the microarray and past literature reports, and were subsequently verified by real-time PCR.

The miRNAs let-7b-5p, miR-1290, miR-34a-5p, and miR-3648 identified in mouthrinse-derived exosomes were significantly upregulated in SS compared with control samples. Among these, the AUC of let-7b-5p, miR-1290, and miR-34a-5p was >0.7 and had good discriminating power. In this context, rheumatoid factor (RF), anti-SS-A, anti-SS-B, and antinuclear (ANA) antibodies are currently applied. However, disease-specific biomarkers have not yet been identified, and their inclusion in diagnostic criteria is inconsistent. The reported sensitivity and specificity are, respectively, 83.7% and 91.5% for anti-SS-A and/or anti-SS-B antibodies, 72.3% and 86.4% for RF, and 72.8% and 80.4% for ANA [5]. The diagnostic performance of the let-7b-5p and miR-1290 combination, calculated using a technique for combined miRNAs and multivariate analysis, was better than that for either miR alone, which had been comparable to extant autoantibody tests. The AUC of the let-7b-5p and miR-1290 combination was 0.856, with 91.7% and 83.3% sensitivity and specificity, respectively. Therefore, the let-7b-5p and miR-1290 combination may be fit to diagnose SS from mouthrinse samples.

Associations between let-7b-5p and SS-A/Ro and SS-B/La antigens in peripheral blood mononuclear cells, minor salivary gland tissues, and salivary gland epithelial cells of SS patients have been identified [31,32]. The MiR let-7b-5p has proven useful as a biomarker of multiple sclerosis [33]. Notably, let-7b targets Hmga2, regulates the function of neural stem cells [34], and is also involved in cell proliferation, the cell cycle, apoptosis, and metabolism through targeting IGF1R [35]. Furthermore, let-7b is associated with *GTF2I*, an SS susceptibility gene [36,37]. As such, let-7b has many predicted targets and is likely to be involved in cell and biological homeostasis, as well as in the pathogenesis of SS, but this notion awaits further investigation.

Previous studies have found that miR-1290 targets *NAT1* and *ALDH4A1* and is involved in lung cancer tumor cell proliferation and infiltration [38]. Among autoimmune diseases, miR-1290 might be specifically involved in celiac disease [39]. The SS susceptibility gene *GTF2I* is associated with miR-1290 [40,41], but data on the involvement of miR-1290 in the onset or pathophysiology of SS are limited; further investigation is warranted to elucidate the biological functions of miR-1290, including its target gene or genes.

MiR-34a-5p is involved in cell apoptosis, bone metabolism, and immune regulation [42], and perhaps also in the immunosuppressive pathway of dendritic cells in the synovial tissues of patients with rheumatoid arthritis [43]. In addition, miR-34a targets *Sirtuin 1* and *Foxp 1*, which are involved in the production of Th17 and follicular T cells [44]. Moreover, miR-34a is upregulated in the synovium, lymph nodes, and spleen in mouse models of rheumatoid arthritis, and hence, inhibiting it improves arthritis [45]. Therefore, targeting miR-34a-5p might also have a powerful impact on SS therapy, although further investigations are needed.

The present study has some limitations. Given the high prevalence of females with this condition, only female patients were chosen for this study. However, since male patients also suffer from SS conditions, it is necessary to study both male and female cases in the future. Many patients had a long history of SS and treatment; whether the results may differ during the early stages of the disease or course of treatment remains unknown. In addition, further investigation is needed to determine whether the identified miRNAs could serve as diagnostic markers for SS or to monitor pathological states. It has also been reported that miRNAs with increased expression levels are involved in other autoimmune diseases. This study did not include other autoimmune diseases (such as RA and SLE) due to the cost and time that such an analysis would require. In the future, other autoimmune diseases should be studied to demonstrate whether the identified miRNAs are SS-specific biomarkers. In this study, the control counterparts of SS patients were healthy subjects, but in future studies, dry mouth patients (non-SS) should also be included as controls. Plus, although saliva can be an effective indicator of local and systemic diseases, the relationship between mouthrinse and saliva is unclear; to learn more, correlations between mouthrinse, saliva, and salivary gland tissue samples should be investigated via exosome profiling. As a final point to note, the results of this study were obtained at a single institution, which limits their general validity; two-step verification in a different cohort is required to further validate our findings.

## 5. Conclusions

Our study identified microRNAs that are upregulated in SS patients from exosomes contained in discharged mouthrinse. Our results indicate that a combination of microRNAs can be used to detect SS patients with relatively high sensitivity and specificity. Using exosome-derived microRNAs from mouthrinse serves as a novel, non-invasive method for sample collection, and these microRNAs have demonstrated potential as biomarkers for SS. It may also be possible to use this method for SS progression monitoring. We plan to analyze SS stage-specific miRNA profiles in future research.

## Figures and Tables

**Figure 1 jpm-12-01483-f001:**
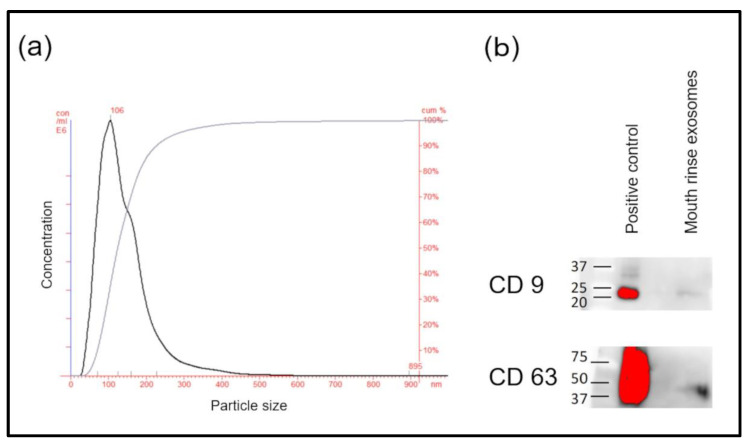
Confirmation of exosome pellets. (**a**) Exosome size distribution analyzed using nanoparticle tracking. (**b**) Western blots of exosome-specific markers, CD9 and CD63.

**Figure 2 jpm-12-01483-f002:**
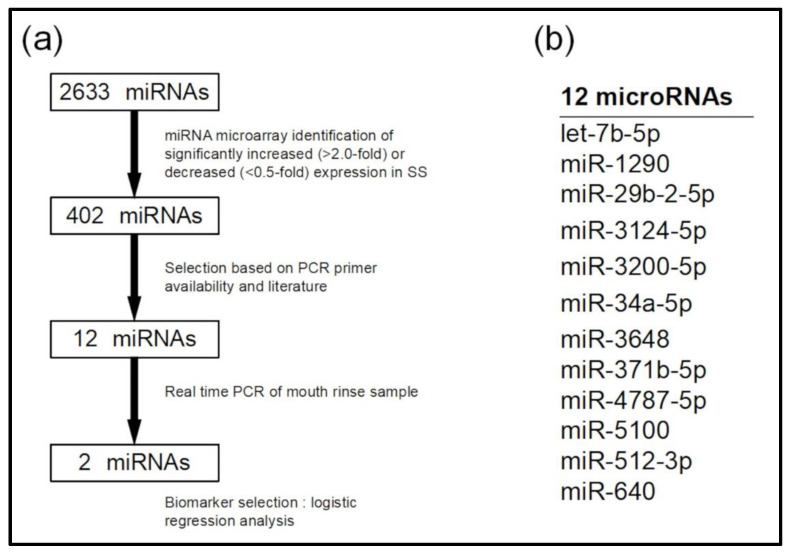
Selection of candidate microRNAs (miRNAs) using microarrays. (**a**) Flow diagram of candidate miRNA detection. (**b**) MicroRNAs validated using real-time PCR.

**Figure 3 jpm-12-01483-f003:**
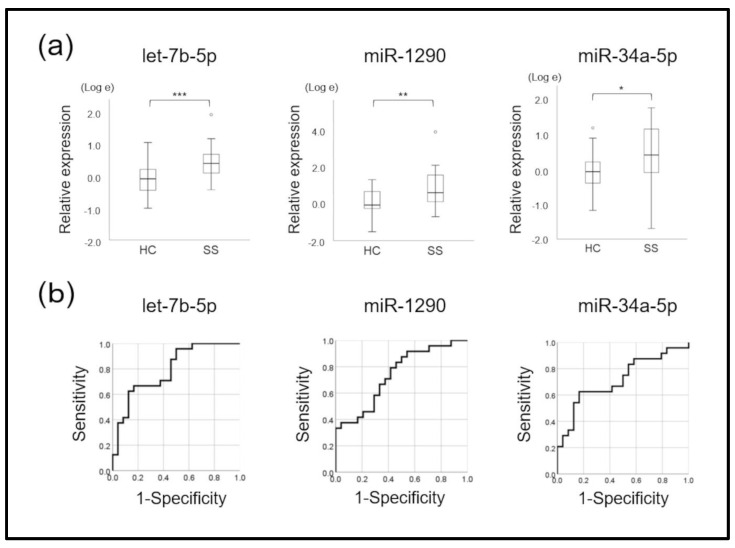
Three microRNAs with significant differences. (**a**) Comparison of normalized signal intensity of three microRNAs (miRNAs) between patients with Sjögren syndrome (SS) and controls (HC). There were significant differences in signal intensity between miRNAs let-7b-5p, miR-1290, and miR-34a-5p (*p* < 0.05, *t*-tests). * *p* < 0.05, ** *p* < 0.01, *** *p* < 0.001. (**b**) The areas under ROC curves (AUCs) were higher for the miRNAs let-7b-5p, miR-1290, and miR-34a-5p than the other six miRNAs, which could not significantly discriminate Sjögren syndrome (SS) from controls. * *p* < 0.05, ** *p* < 0.01, *** *p* < 0.001.

**Figure 4 jpm-12-01483-f004:**
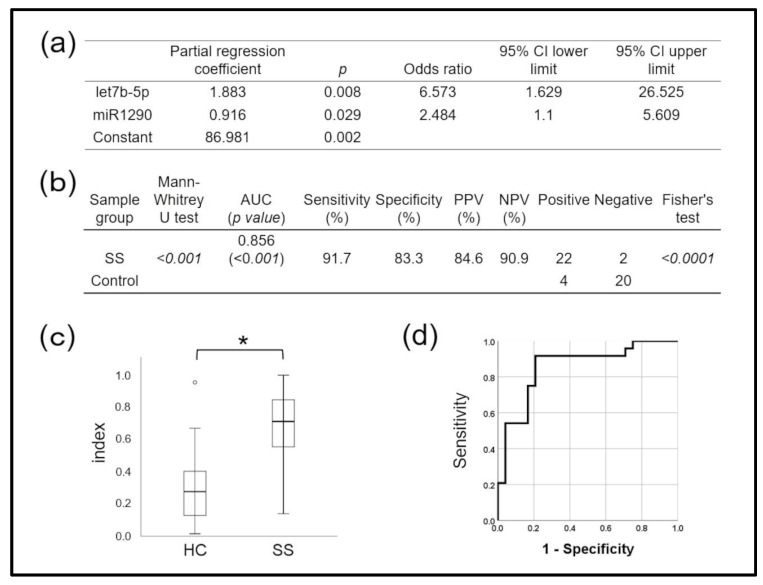
Diagnostic performance of two combined microRNAs (miRNAs) for Sjögren syndrome (SS), determined using a diagnostic (miRNA) index and calculated using logistic regression analysis. (**a**) Results of logistic regression analysis. Model’s X^2^ test *p* < 0.001, Hosmer–Lemeshow test *p* = 0.836, disease discrimination rate = 81.2%. Prediction formula = 1/(1 + exp (−1 × score)). Score = 86.981 + (1.883 × let7b-5p) + (0.916 × miR1290). (**b**) Ability of miRNA index to diagnose Sjögren syndrome (SS) based on cut-off value of ROC curve. (**c**) There was a significantly higher miRNA index in SS patients than in controls (* *p* < 0.001, Mann–Whitney *U* tests). (**d**) Diagnostic performance based on AUC, 0.856; sensitivity, 91.7%; specificity, 83.3%. AUC, area under the ROC curve; NPV, negative predictive value; PPV, positive predictive value; ROC, receiver operating characteristics.

**Table 1 jpm-12-01483-t001:** Clinical characteristics of study participants.

	SS Group	HC Group	
	*n* = 24	*n* = 24	*p*-Value
Characteristic			
Female, *n* (%)	24 (100)	24 (100)	
Age, mean (range)	63.8 (35–86)	63.5 (41–84)	ns
Smoker, *n* (%)	0 (0)	1 (4.1)	ns
Clinical symptoms			
Disease duration, median (range), months	59.5 (13–204)	NA	
Anti-SS A positive, *n* (%)	18 (75)	NA	
Anti-SS B positive, *n* (%)	10 (42)	NA	
Saxon test, median (range), g	1.65 (0.2–6.1)	NA	
Treatment			
M3R agonist, *n* (%)	15 (62.5)	0	
Corticosteroid, *n* (%)	3 (12.5)	0	
Immunosuppressive drug, *n* (%)	0 (0)	0	

Footnote: Ages were compared using *t*-test, and smokers were compared using Fisher’s exact test. SS: Primary Sjögren syndrome; HC: Healthy control; ns: No significant difference; NA: Not applicable.

**Table 2 jpm-12-01483-t002:** Diagnostic performance of miRNAs for Sjögren syndrome (SS) detection using cut-off values.

MicroRNA	Group	AUC (*p*-Value)	Sensitivity (%)	Specificity (%)	PPV (%)	NPV (%)	Positive (*n*)	Negative (*n*)	Fisher’s Test (*p*-Value)
let-7b-5p	SS	0.793 (*<0.001*)	62.5	83.3	78.9	69	15	9	*0.003*
	HC						4	20	
miR-1290	SS	0.74 (*0.004*)	83.3	54.2	64.5	76.5	20	4	*0.015*
	HC						11	13	
miR-34a-5p	SS	0.751 (*0.011*)	62.5	83.3	78.9	69	15	9	*0.003*
	HC						4	20	
miR-3648	SS	0.653 (*0.07*)	70.8	58.3	63	66.7	17	7	*0.08*
	HC						10	14	
miR-3200-5p	SS	0.582 (*0.332*)	45.8	79.2	68.8	59.4	11	13	*0.125*
	HC						5	19	
miR-3124-5p	SS	0.545 *(0.592*)	37.5	62.5	50	50	9	15	*1* *.00*
	HC						9	15	
miR-4787-5p	SS	0.531 *(0.711*)	58.3	45.8	51.9	52.4	14	10	*1* *.00*
	HC						13	11	
miR-5100	SS	0.563 (*0.488*)	70.8	33.3	51.5	53.3	17	7	*1* *.00*
	HC						16	8	
miR-512-3p	SS	0.543 (*0.606*)	87.5	33.3	56.8	72.7	21	3	*0.168*
	HC						16	8	

Footnote: AUC: area under the curve; PPV: positive predictive value; NPV: negative predictive value; Fisher’s test: Fisher’s exact test.

## Data Availability

The data presented in this study are available on request from the corresponding author. The data are not publicly available due to privacy and ethical restrictions.

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
