# Peer review of "Exosome-Derived microRNAs from Mouthrinse Have the Potential to Be Novel Biomarkers for Sjögren Syndrome"

_jpm, 2022, doi:10.3390/jpm12091483_

Round 1

Reviewer 1 Report

Overall a nicely presented and executed research project with good scientific soundness. 

1. Pls provide a table for inclusion and exclusion criteria.

2. Also , it would be better if the authors can provide a picture of the guideline for selection criteria of Japanese ministry etc.

Reviewer 2 Report

This manuscript presents experimental work towards developing a non-invasive method for diagnosing Sjogren syndrome as an alternative to the more invasive methods currently available. Evidence is presented in support of this method, based on collecting mouthrinse samples from patients from which the increased expression ratios of four miRNAs, among these miR-1290 and let-7b-5p especially, served as predictors for the presence or absence of the medical condition, the method being limited to patients who had been experiencing symptoms for a fairly long time and had already received some form of treatment thereof. I believe this work is a good stepping stone in the direction of SS diagnosis and its limitations are fairly outlined at the end of the manuscript. Thus, it may be accepted for publication in JPM.

A few modifications and clarifications are still needed:

1. Why were only female subjects chosen in this study? Although female patients are more frequently encountered with this medical condition, male patients do exist as well, so this should at least count as another limitation of the study that should be added to the summary at the end of the Discussion section.

2. Line 163: I believe you meant to write p less or equal to 0.05, not just equal as it reads currently.

3. The text within Figure 3 especially and in part also Figure 4 is of low resolution, making it difficult to read. Please address this! 

4. Reference formatting: some journal names are abbreviated, while others are not. Please reformat in a consistent manner throughout the entire bibliography! 
